# Effect of Active Packaging Material Fortified with Clove Essential Oil on Fungal Growth and Post-Harvest Quality Changes in Table Grape during Cold Storage

**DOI:** 10.3390/polym13193445

**Published:** 2021-10-08

**Authors:** Siriporn Luesuwan, Matchima Naradisorn, Khursheed Ahmad Shiekh, Pornchai Rachtanapun, Wirongrong Tongdeesoontorn

**Affiliations:** 1School of Agro-Industry, Mae Fah Luang University, 333 Moo 1 Tasud, Chiang Rai 57100, Thailand; siriporn.lue@mfu.ac.th (S.L.); matchima@mfu.ac.th (M.N.); khursheed.research@mfu.ac.th (K.A.S.); 2Research Group of Postharvest Technology, Mae Fah Luang University, 333 Moo 1 Tasud, Chiang Rai 57100, Thailand; 3Research Group of Innovative Food Packaging and Biomaterials Unit, Mae Fah Luang University, 333 Moo 1 Tasud, Chiang Rai 57100, Thailand; 4School of Agro-Industry, Faculty of Agro-Industry, Chiang Mai University, Chiang Mai 50100, Thailand; pornchai.r@cmu.ac.th; 5The Cluster of Agro Bio-Circular-Green Industry (Agro BCG), Chiang Mai University, Chiang Mai 50100, Thailand; 6Center of Excellence in Materials Science and Technology, Chiang Mai University, Chiang Mai 50200, Thailand

**Keywords:** *Aspergillus* sp., deterioration, weight loss, antifungal clove oil, polyvinyl alcohol film, eco-friendly, grape berry, shelf life

## Abstract

Fungal growth in table grapes (*Vitis vinifera* cv. beauty seedless) is triggered by *Botrytis cinerea*, *Penicillium* sp., *Aspergillus* sp., and *Rhizopus stolonifera* during post-harvest storage. Due to the safety aspects, this research aimed to develop antifungal packaging embedded with essential oils (EOs) to alleviate the fungal decay of table grapes (TG). The various levels of EOs (0.5–5%, *v*/*v*) from clove, cinnamon, thyme, peppermint, lemon, bergamot, ginger, spearmint, and lemongrass were tested against *Aspergillus* sp. The results attained in radial growth, disk diffusion method, minimal inhibitory concentration, and minimal fungicidal concentration revealed that 1% clove essential oil (CEO) showed higher efficacy against *Aspergillus* sp. compared to the untreated control and other treatments. CEO at the 1% level exhibited a pleasant odor intensity in TG than the other EOs. The active polyvinyl alcohol (7% PVA) film with 1% CEO resulted in lower weight loss, disease severity, and TG berry drop than the control and other treated samples. Additionally, the acceptance score in the TG sample wrapped with a PVA film containing 1% CEO was augmented. Therefore, the PVA film with 1% CEO retarded the fungal growth and prolonged the shelf life of TG during storage of 21 days at 13 °C and 75% relative humidity (RH).

## 1. Introduction

The table grape (*Vitis vinifera* L. cv. ‘Beauty seedless’) is an important economic crop that suffers severe quality losses due to different spoilage and pathogenic microbial species during post-harvest storage [1]. The deterioration of table grapes (TG) is catalyzed by the different types of microorganisms such as bacteria (*Gluconobacter* sp. and *Acetobacter* sp.), yeasts (*Zygosaccharomyces* sp.), and molds (*Botrytis cinerea, Penicillium* sp., *Aspergillus* sp., and *Rhizopus stolonifera*) [2,3]. Additionally, pathogenic fungi in grapes such as *Botrytis cinerea*, *Aspergillus* sp., and *Penicillium* sp. allow the development of aerial mycelium to spread rapidly to adjacent berries with severe economic repercussions [4]. Besides microbial deterioration, grapes are perishable non-climacteric fresh food commodities with several other post-harvest storage problems such as loss of firmness, berry drop, stem discoloration, and desiccation [2]. Seedless grape cultivars may be more economical to farmers, supermarkets, and export markets if wrapped with a suitable bioactive packaging material [5].

Over recent decades, several chemical methods have been used to preserve TG. Sulfur dioxide (SO_2_) fumigation has been practiced commonly to prevent the microbial deterioration of TG. Although SO_2_ fumes may retard the fungal growth, sulfite residues may be toxic to the health of consumers [2,6]. Due to ethical concerns, the Food and Drug Administration (FDA) has set the maximum tolerance for sulfite residues at 10 ppm generated by SO_2_ pads containing sodium metabisulfite (Na_2_S_2_O_5_) enclosed in paper and plastic sheets [6,7]. Although commercial SO_2_ pads may prevent fungal growth to some extent, excess amounts may damage the post-harvest quality of TG. Degradation of quality attributes by a high concentration of SO_2_ may cause bleaching, shrinkage of grape berries, early browning of the rachis, cracking, fruit injury, unpleasant aftertaste, and food allergies [6]. 

Essential oils (EOs) are natural byproducts from plants, GRAS (Generally Recognized as Safe) by the US FDA, having antioxidant, antimicrobial, or antifungal properties [8,9]. EOs have been extracted from different types of plants mainly from the leaves, stems, flowers, seeds, branches, roots, buds, and bark of the plant [9]. EOs extracted from plants are free of toxicity and eco-friendly and may be an efficient alternative to chemical fungicides [10]. The active compounds of EOs are rich in volatile bioactive components mainly constituted by secondary metabolites such as aldehydes, fatty acids, phenols, ketones, esters, and alcohols and exhibit several nutraceutical properties [11]. The use of EOs to extend the shelf life of fruits has gained tremendous interest because of the promising health benefits [12]. Clove (*Syzygium aromaticum*) essential oil (CEO) has been ranked as a natural food additive by the Food and Drug Administration (FDA, USA) [13]. CEOs can be extracted from the buds, leaves, and stems of plants which differ in their color, flavor, and chemical composition. The best-quality CEO is obtained from the buds. GC-MS analysis detected the various components in CEO such as oxygenated monoterpenes (84.31%), sesquiterpene hydrocarbons (15.45%), and oxygenated sesquiterpenes in which eugenol (69.7%), eugenyl acetate (14.4%), and β-Caryophyllene (12.2%) were major constituents [14]. Clove essential oil has been reported to show antifungal activity against human pathogenic fungi (*Trichophyton rubrum*, *Trichophyton mentagrophytes*, and *Epidermophyton flocosum*), *Candida* sp., and *Aspergillus* sp. [15,16,17].

Polyvinyl alcohol (PVA), as one of the biodegradable polymers, has been brought into focus due to its outstanding biocompatibility, film formation, and gas barrier performance in fresh fruit packaging [18]. PVA alone, being short of antimicrobial properties, limits the wider application in food packaging. A PVA doped with 1% ε-polylysine packaging film could effectively inhibit pericarp browning and pulp breakdown in longan, consequently increasing the rate of commercial acceptability [19]. A PVA-supplemented nano-silver composite film was tested on *Aspergillus niger*, which showed higher antimicrobial efficacy under in vitro conditions (zone of inhibition 14.4 mm and minimum bactericidal concentration of 75 ppm) and remarkably extended the shelf life of packaged grapes [20]. PVA (10%, *w*/*v*) and chitosan (25%, *w*/*v*) were fabricated into a bilayer film with excellent barrier and antimicrobial properties, employed in the packaging of strawberries to extend the shelf life [21]. In general, biopolymer-based films or bio-composite sheets have been recommended to be eco-friendly compared to non-biodegradable plastics [22,23,24].

Application of biodegradable packaging films and edible coatings fortified with EOs has developed the concept of post-harvest preservation of fresh produce [10]. Active packaging materials for grapes supplemented with EOs have been evidenced to develop a protective atmosphere rich in volatile antimicrobial or antioxidant components against microbial spoilage in TG [25]. Food packaging films from PVA and chitosan enriched with cinnamon EOs were reported to have excellent antimicrobial properties [26]. PVA-loaded CEO nano-size capsules exhibited antifungal properties [27]. Nevertheless, there is no information about the preparation of PVA films loaded with CEO on the retardation of fungal growth and quality changes in TG. Therefore, the current study was conducted to screen the desirable level of EOs based on antimicrobial characterization under in vitro conditions. The impact of a potential CEO-fortified antifungal packaging material was also investigated on the fungal growth and post-harvest quality changes in TG during storage of 30 days at 13 °C and 75% relative humidity (RH).

## 2. Materials and Methods

### 2.1. Materials 

Chemicals and microbial media used in the experiments were of analytical grade. Dimethyl sulfoxide (DMSO) (RCI Labscan Limited, Bangkok, Thailand), Tween-80, potato dextrose agar (PDA), and halloysite nano clay were procured from Sigma-Aldrich (St. Louis, MO, USA). The fungal culture, typically *Aspergillus* sp., was isolated by the Thailand Bioresource Research Center (TBRC). Sodium hypochlorite (10%) was purchased from KrungthepChemi (Bangkok, Thailand). All the commercial-grade essential oils (EOs) such as lemongrass, spearmint, ginger, bergamot, lemon, peppermint, thyme, cinnamon, and clove were obtained from J.A.G.A.T. Aroma oils distillation, Namsiang Co., Ltd. (Bangkok, Thailand). Polyvinyl alcohol (PVA) and oriented polypropylene (OPP) films were bought from Loba Chemie PVT. Ltd. (Maharashtra, India). Sterile absorbent food-grade pads (25 mL capacity) were purchased from Dry Square Co., Ltd. (Bangkok, Thailand). 

### 2.2. Effect of Different Levels of EOs on Aspergillus sp.

The different types of EOs from clove, cinnamon, thyme, peppermint, lemon, bergamot, ginger, spearmint, and lemongrass were screened against *Aspergillus* sp. All the EOs were preprepared in dimethyl sulfoxide (DMSO) at various levels of 0.5, 1, 2, and 5% (*v*/*v*). The various levels of EOs were subjected to in vitro microbial analyses. 

#### 2.2.1. Radial Growth and Disk Diffusion Method

Different levels of EOs (0.5, 1, 2, and 5 % *v*/*v*) were prepared, and 100 µL of each level of EOs was mixed with 20 mL of sterile potato dextrose agar (PDA). The mixtures of EOs and PDA were allowed to solidify on Petri dishes. A circular spot of 3 mm in diameter was made using a sterile cork-borer at the center of each Petri dish. *Aspergillus* sp. Suspension of 10 µL (10^6^ CFU mL^−1^) was dropped on the circular spot of Petri dishes with different levels of EOs. Control was prepared by the addition of 10 µL of sterile distilled water instead of EOs on the spot of the inoculated Petri dish. Inoculated Petri dishes were sealed using parafilm to avoid contamination. The plates were incubated for 7 days at 25 ± 2 °C, and the radial growth was measured from the developed mycelium on the plates [28]. Disk diffusion was analyzed by placing a 6 mm sterile paper disk in the center of the inoculated plate with *Aspergillus* sp. suspension (100 µL). Diluted concentrations of EOs (10 µL) were dropped on paper disks and incubated. The inhibition zones of different levels of EOs were measured based on the growth of mycelium [29]. 

#### 2.2.2. Minimum Inhibitory Concentration (MIC) and Minimum Fungicidal Concentration (MFC)

Spore suspension of *Aspergillus* sp. was prepared in potato dextrose broth (PDB) by serial dilution (10^5^ CFU mL^−1^). To 100 µL of spore suspension, 10 µL was added from all the EO samples in the wells of a sterile microplate. Microplates were incubated at 25 ± 2 °C for 24 h. Absorbance was measured using a microplate reader for the determination of MIC of EOs [30]. MFC was analyzed by spreading 100 µL from the selected wells of the MIC microplate on the PDA plate. All the plates were incubated at 25 ± 2 °C for 48 h to determine MFC [30].

### 2.3. Procurement and Preparation of TG Samples 

Fully ripened TG (*Vitis vinifera* cv. Beauty seedless) were harvested from PB Valley Chiang Rai orchard located in the north of Thailand. Good Agricultural Practices (GAP) were followed until the harvesting time. Freshly harvested grapes ranged 14–18 Brix in total soluble solids (TSS), certified by the Department of Agriculture, Thailand. Fresh TG were transported to the Postharvest Technology and Packaging Laboratory, Mae Fah Luang University, Chiang Rai, Thailand. Selection of TG was conducted based on uniform size, color, appearance, and absence of mechanical bruises [4]. After selection, TG were immersed in 200 ppm sodium hypochlorite for 2 min to remove the extraneous material and microbial contamination. Washed TG samples were dried in a biosafety cabinet to avoid cross-contamination until use.

### 2.4. Evaluation of Odor Intensity in TG Treated with Different EOs

EOs at the 1% level were analyzed for odor intensity in which 10 trained panelists were recruited from the Department of Post-harvest Technology, Mae Fah Luang University, Chiang Rai, Thailand, aged between 25 and 32 years. EOs (1 mL) at the 1% level were poured in all the sealable cups followed by the addition of a TG sample (10 berries, 50 g). All the cups were sealed airtight and incubated at 13 °C for 24 h. After incubation, all the sample cups were kept at room temperature for 2 h before being presented to the panelists. The panelists were asked to open the sealed cups and sniff the headspace to determine the odor intensity of the grape berries [31]. All the grape berry samples treated with different EOs were scored between 0 (none) and 4 (extremely strong odor) [32].

### 2.5. Preparation of Antifungal Active Packaging for Quality Preservation of TG

Clove essential oil (1% CEO) was supplemented based on the pleasant odor intensity score by the trained panelists as an active antifungal ingredient in different packaging materials. Firstly, sterile absorbent pads were injected with 25 mL of 1% CEO and placed in a zip-lock bag at 25 °C. Secondly, 7% polyvinyl alcohol (PVA) was dissolved in distilled water, heated on a magnetic stirrer set at 98 ± 2 °C to obtain film forming solution (FFS), and cooled down at room temperature. Then, FFS was combined with 20% of Tween 80 and 1% CEO and was divided into 2 portions. The first portion of FFS constituted PVA and 1% CEO only, while the second portion of FFS (PVA and 1% CEO) was further combined with 1% halloysite clay. Finally, FFSs containing PVA and 1% CEO without and with 1% halloysite clay were casted on the orientated polypropylene film (OPP). The films were dried at room temperature for 24 h, manually peeled off, and conditioned in an environmental chamber (WTB Binder, Tattling, Tuttlingen, Germany) at 25 °C and 50% relative humidity for 24 h before use [33].

#### Application of Antifungal Active Packaging on TG during Post-Harvest Cold Storage

Washed TG samples (300 ± 20 g/treatment) free of contamination were packed in polypropylene (PP) bags with 6–8 holes of diameter 1 cm and placed in a corrugated box. Different treatments of TG were prepared such as control (without any treatment), CEO1-Pad (absorbent pad injected with 25 mL of 1% CEO), COM-SO_2_-Pad (commercial pad of sodium metabisulfite 98%), CEO1-HC-Film (7% PVA film fortified with 1% CEO and 1% halloysite clay), and CEO1-Film (7% PVA film fortified with 1% CEO). All the pad and film treatments were placed on the grape samples packaged in PP bags kept in corrugated boxes. Boxes were sealed and replications (n = 6) were carried out for each day of analysis with an interval of 3 days up to 30 days, stored at 13 °C, 75%RH.

### 2.6. Effect of Antifungal Active Packaging on Post-Harvest Quality Losses of TG

#### 2.6.1. Disease Severity and Grape Berry Drop 

Disease severity was analyzed following a 6-point empirical scale (0 = 0% fruit surface infected; 1 = 1–20% fruit surface infected; 2 = 21–40% fruit surface infected; 3 = 41–60% fruit surface infected; 4 = 61–80% fruit surface infected; 5 = 81% or more surface of fruit infected and showing sporulation) [34,35]. The fruit drop of TG was analyzed at every 3-day interval of storage up to 30 days [35]. 

#### 2.6.2. Weight Loss (%)

The weight loss during post-harvest storage was determined in various replications (n = 6) at every 3-day interval up to 30 days of storage [5].
(1)$$Weight\;loss\;\left(\%\right)=\left[\frac{\left(Mi-Ms\right)}{Mi}\right]\times 100$$
where *Mi* and *Ms* are the initial and final weights of the samples, respectively. 

#### 2.6.3. Acceptance Score

Fifty untrained panelists comprising 27 males and 23 females aged 25–32 years were recruited from the School of Agro-Industry, Mae Fah Luang University, Chiang Rai, Thailand. All the panelists were asked to evaluate the appearance, firmness, odor, and overall acceptability of TG samples without and with active antifungal packaging material, using a 9-point hedonic scale at day 0 and day 21 of storage at 13 °C [36,37].

### 2.7. Statistical Analysis

Analysis of variance (ANOVA) and Duncan’s multiple range test were performed using a statistical program, SPSS (Chicago, IL, USA) v. 10.0. Samples were analyzed at a level of significance of *p* < 0.05 for all the parameters.

## 3. Results and Discussion

### 3.1. Impact of EOs at Different Levels on the Radial Growth, Disk Diffusion Method, MIC, and MFC of Aspergillus sp.

The efficacy of different levels of EOs in fungal growth inhibition, displayed on the PDA plates inoculated with *Aspergillus* sp., is presented in Table 1. Radial growth in the control (without any treatment) was higher than the plates treated with different levels of EOs (*p* < 0.05). Inoculated plates treated with EOs at various levels of 0.5, 1, 2, and 5% (*v*/*v*) showed decreases in the radial growth of *Aspergillus* sp. compared to the untreated control sample (*p* < 0.05). Additionally, the results of radial growth attained in lemongrass, thyme, cinnamon, and clove EOs at different levels are lower than the control and other EOs (Table 1). However, CEO had the lowest radial growth of fungal mycelium in comparison to the control and other treated plates (*p* < 0.05). Moreover, the radial growth was observed to be concentration-dependent in CEO, such that CEO above the 1% level and up to the 5% level showed higher radial growth inhibition. The higher efficacy of radial growth inhibition of CEO at the 5% level might be correlated with the higher amount of active chemical constituents that exhibit antifungal activity [14]. A combination of EOs from clove and mustard at different concentrations of 11.57 μL/L_air_ and 1.93 μL/L_air_ was reported to inhibit *Botrytis cinerea* under in vitro conditions isolated from strawberries, respectively [10].

The values of the inhibition zone of *Aspergillus* sp. plates treated with different levels of EOs via the disk diffusion method are shown in Table 1. The control (without EOs) showed no inhibition zones of fungal growth. However, plates with EOs at the 0.5–5% level showed retarded growth measured by the diameter of clear spots or zone of inhibition on the inoculated plates (Table 1). The slight increment in the inhibition zones was marked in a dose-dependent manner among all the EOs excluding CEO (*p* > 0.05). Additionally, the diameter of the inhibition zone was notably higher in CEO for all the tested levels compared to the other plates treated with different EOs (*p* < 0.05). It was evidenced that the wideness of the inhibition zones or clear spots were higher in plates treated with 1% to 5% CEO. Moreover, the presence of active antifungal components in 1–5% CEO might be the potential cause of the inhibition zones displayed on the *Aspergillus* sp. inoculated plates. The results are in line with the inhibition zone of fungal *Aspergillus* sp. obtained by the treatment of oregano and clove EOs [38].

The in vitro MIC and MFC values of the different levels of EOs attained in the microplate wells inoculated with serially diluted *Aspergillus* sp. culture are presented in Table 2. Lower values of MIC were obtained in lemongrass, thyme, and clove EOs than the other EOs (*p* < 0.05). The MIC values of the aforementioned EOs at the 1–5% level could inhibit the growth of mycelium formation in a dose-dependent pattern. However, MIC values were the lowest in 5% CEO compared to the other EOs (*p* < 0.05). Similarly, MFC values of EOs were half of the concentrations of MIC. Moreover, the decreases in MFC values for 5% CEO depict that the lowest concentration of CEO was employed for effective inhibition of *Aspergillus* sp. It was reported in a study that different EOs from oregano, clove, thyme, lavender, clary sage, and arborvitae at various concentrations of 75, 50, 25, 10, and 5% (*w*/*v*) were employed against *Alternaria alternata* and *Aspergillus fumigatus*. The mycelial growth of the tested fungal strains was retarded by oregano and clove, having MICs of 0.01% and 0.025% and MFCs of 0.025% and 0.05%, respectively [38]. A comparative study of thyme, oregano, and lemon EOs reported that *Botrytis cinerea*, *Penicillium italicum*, and *Penicillium digitatum* were inhibited by thyme EO more effectively [28]. Additionally, EO from cloves during the in vitro vapor phase with an MIC value of 92.56 µL/L_air_ was documented to inhibit the mold growth triggered by *B. cinerea* [10]. As a matter of fact, plant EOs containing volatile polyphenolic compounds such as terpenes, phenolics, aldehydes, and alcohols that exhibit antimicrobial or fungicidal activity might alter or inhibit the growth mechanism of food-borne pathogenic fungal species [39]. 

### 3.2. Evaluation of Odor Intensity of EOs in TG 

The odor intensity scores of different EOs at the 1% level in TG are shown in Figure 1. Desirable odor intensity scores sniffed by the trained panelists on the headspace of cups were given to the TG berries treated with clove, bergamot, cinnamon, lemon, and thyme EOs incubated at 13 °C for 24 h (*p* < 0.05). The highest scores attained for spearmint and peppermint EOs indicate a strong undesirable odor in the TG berries. Conversely, the lowest odor intensity score obtained with 1% CEO-treated TG was referred to as the most pleasant odor amongst all the EOs (*p* < 0.05). CEO at 1% was a more suitable concentration of CEO evaluated on the basis of the odor intensity score for the preparation of an antifungal packaging material without compromising the sensorial properties of TG. From previous studies, CEO was reported to be rich in two major flavor compounds, eugenol and eugenol acetate, that constitute 86% of the analyzed compounds by GC-MS in CEO [40,41]. 

### 3.3. Effect of Antifungal Packaging Materials Supplemented with CEO on Quality Changes in TG during Storage

#### 3.3.1. Disease Severity

The growth of fungal mycelium monitored by disease severity on TG without and with CEO packaging treatments is shown in Figure 2A. Fungal growth was triggered in the control at day 3, while in the CEO1-Pad sample, mycelial growth was initiated from day 6 of storage at 13 °C (*p* < 0.05). Simultaneously, the fungal growth in the COM-SO_2_-Pad sample more likely coincided with the CEO1-Pad sample and marked a visible difference from the TG samples wrapped with active antifungal films fortified with 1% CEO (*p* < 0.05). CEO1-HC-Film and CEO1-Film samples exhibited slight fungal growth from day 9, compared to the control and pad treatments (*p* < 0.05). However, the lowest severity of disease was visualized in the CEO1-Film sample, compared to the rest of the untreated and treated TG during the storage of 21 days at 13 °C. Although commercial SO_2_ pads generated fumes that could aid in the decrease in mold growth to some degree in TG, they have been reported with several constraints. SO_2_ fumigation in seedless TG has been documented with the damage of berries detected by hairline cracks, bleaching, and early browning of the rachis [42,43]. Nonetheless, the CEO1-Film sample was the most effective antifungal packaging material that might release the volatile fungicidal components of CEO due to the large surface area of the film. Furthermore, the volatile CEO components could reside in the voids between the berries of TG to prevent cluster growth of fungal mycelium. Photographs of the control and treated TG samples for visualization of fungal growth are presented in Figure 2B. Thus, CEO contains major components of antifungal agents that have been employed in different techniques of packaging for the prevention of TG decay [44,45].

#### 3.3.2. Weight Loss 

The percent weight loss values of TG treated with different packaging materials fortified with 1% CEO are presented in Figure 3A. Within the first 6 days of storage, all the samples displayed fresh-like characteristics without any marked loss in weight (*p* < 0.05). As the storage proceeded, the control (without any treatment, packaged in PP bags) showed the highest weight loss compared to the treated TG samples during the 30 days of storage (*p* < 0.05). The ascending order of weight loss was observed to be CEO1-Pad > COM-SO_2_-Pad > CEO1-HC-Film > CEO1-Film, not considering the control sample (*p* < 0.05), although the COM-SO_2_-Pad sample showed reduced weight loss in comparison with the control and CEO1-Pad samples (*p* < 0.05). Nevertheless, the lowest weight loss values were measured in the CEO1-Film sample. The results obtained in the weight loss of TG with different packaging treatments are in line with the results of disease severity as evidenced in Figure 2A. The reports of increases in percent weight loss were correlated with the fungal and mold decay of TG [4].

#### 3.3.3. Berry Drop 

The percent of berry drop during cold storage of the control and treated samples of TG is presented in Figure 3B. Berry drop in the control and treated samples was evidenced from day 15 of storage (*p* < 0.05), except for the CEO1-Film sample in which a slight increment was observed at day 18. The control ranked highest in berry drop followed by the descending order of CEO1-Pad, COM-SO_2_-Pad, and CEO1-HC-Film in comparison with the lower berry drop of the CEO1-Film sample at 21 days of storage at 13 °C (*p* < 0.05). The findings of berry drop in the CEO1-Film sample are in line with the disease severity and weight loss shown, respectively, in Figure 2A and Figure 3A. Additionally, the detachment of berries might also be related to the higher fungal disease severity causing interference in the abscission zones connecting the berry and pedicel part of the grape rachis. Therefore, the main mechanism of berry detachment during storage could be correlated with the berry–pedicel indentation and the gradual extension of phloem vessels and pith abscission layer, creating intercellular cavities, leading to berry drop [46].

#### 3.3.4. Sensory Acceptance 

TG samples without and with CEO packaging treatments were analyzed for scores based on appearance, color, odor, and overall acceptability (Table 3). At day 0 of storage, untrained panelists (n = 50) were employed to assess acceptance scores for the control, CEO1-Pad, COM-SO_2_-Pad, CEO1-HC-Film, and CEO1-Film samples. The control and treated samples exhibited the highest score on the first day of storage at 13 °C (*p* < 0.05). The results depict the freshness and glossy appearance of TG berries without any disease or desiccation. With the advancement of the storage time, a gradual increase in physical and microbial quality changes, measured by disease severity, weight loss, and berry drop, degraded the sensorial quality of TG samples (Figure 2 and Figure 3). The control was scored the lowest due to excessive fungal growth (*p* < 0.05), compared to the CEO1-Pad and COM-SO_2_-Pad samples, at day 21 of cold storage. The mist-like appearance of fungal mycelium was quite visible all over the control, CEO1-Pad, and COM-SO_2_-Pad samples.

Additionally, CEO1-HC-Film also exhibited higher scores than those of the aforementioned samples (*p* < 0.05), but slight haziness was also visible on the grapes with less glossiness. This might be due to the slow release of CEO antifungal agents trapped by the halloysite clay particle in the PVA film. Eventually, CEO1-Film samples marked the highest scores for all the tested attributes at 21 days of cold storage. The findings from previous studies state that minimal moisture loss may also lead to browning, desiccation, and wilting, thereby affecting the sensory attributes of TG [47]. Furthermore, EOs from lemon, orange, and mandarin were reported to retard the physicochemical quality changes and microbial growth (molds and yeasts) to safeguard the sensory quality of strawberries during 18 days of cold storage [48]. 

## 4. Conclusions

CEO at the 1–5% level inhibited the growth of *Aspergillus* sp. in a dose-dependent pattern. Lower radial growth and higher inhibition zones were attained with 5% CEO. Lower MIC and MFC values against *Aspergillus* sp. were also attained with 5% CEO. Additionally, the most desirable odor intensity score was assigned to 1% CEO-treated TG berries incubated at 13 °C for 24 h. Thereafter, the control sample (TG without any treatment) lasted only up to 6 days of cold storage at 13 °C. Quality deterioration analyzed by disease severity, weight loss, berry drop, and acceptance score was higher in the CEO1-Pad, COM-SO_2_-Pad, and CEO1-HC-Film samples during 21 days of storage. However, the lower post-harvest quality changes and higher acceptance score prolonged the shelf life of TG in the CEO1-Film sample for up to 21 days of storage at 13 °C. Therefore, CEO (1%)-fortified PVA (7%) active packaging films could be a potential alternative for minimizing post-harvest fungal decay and quality losses in TG. 

## Figures and Tables

**Figure 1 polymers-13-03445-f001:**
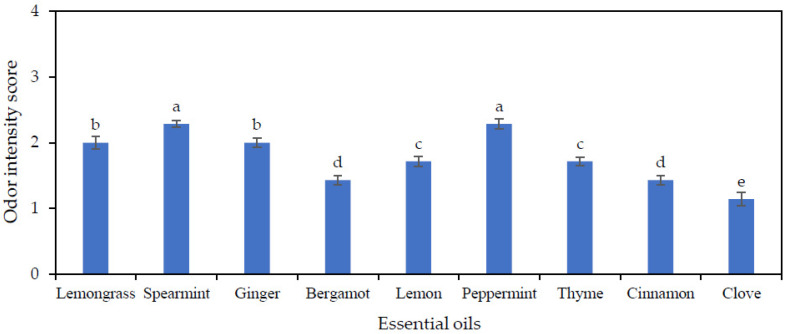
Screening of different essential oils (EOs) based on odor intensity score in table grapes. Values are mean ± standard deviation (n = 10). Different lowercase letters on the bars indicate a significant difference (*p* < 0.05).

**Figure 2 polymers-13-03445-f002:**
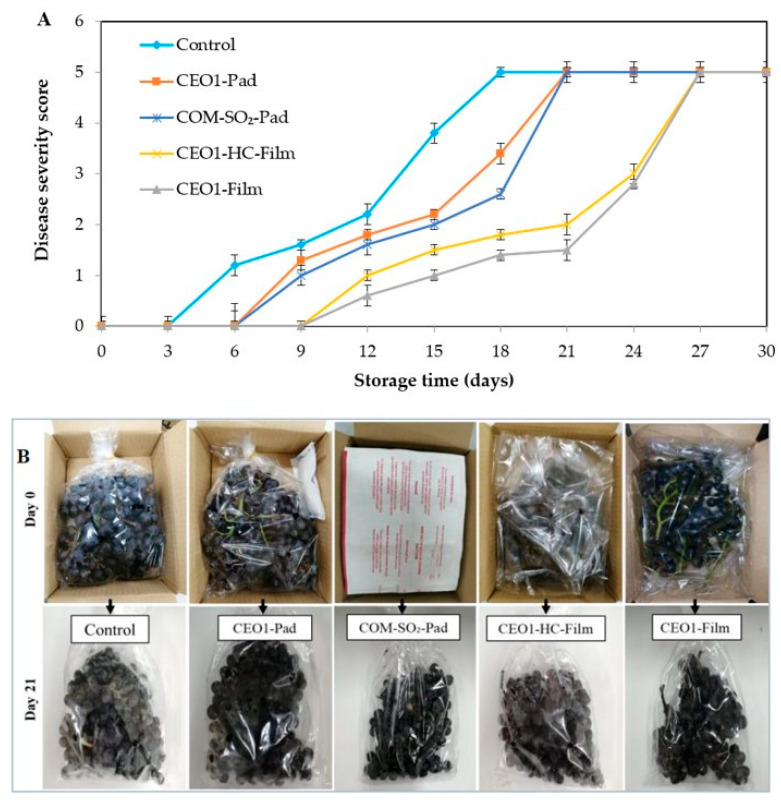
Disease severity (**A**) and photographs (**B**) of TG without and with CEO antifungal packaging. Values are mean ± standard deviation (n = 10). TG: table grape; CEO: clove essential oil; Control: TG without any treatment; CEO1-Pad: TG with 1% CEO pad; COM-SO_2_-Pad: TG with commercial sulfur dioxide pad; CEO1-HC-Film: TG wrapped with a film composed of 1% CEO and halloysite clay; CEO1-Film: TG wrapped with 1% CEO film.

**Figure 3 polymers-13-03445-f003:**
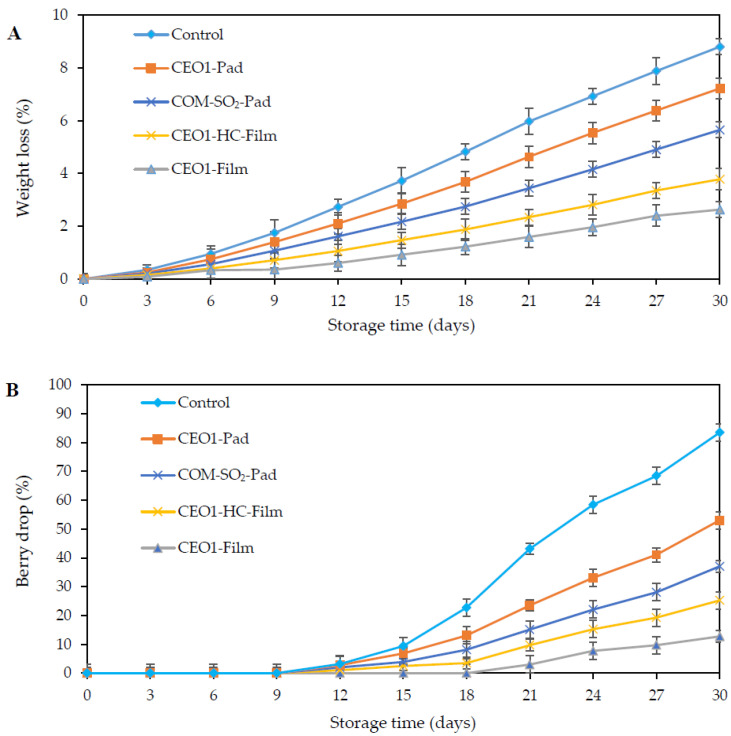
Weight loss (**A**) and berry drop (**B**) of TG without and with CEO antifungal packaging. Values are mean ± standard deviation (n = 6). See Figure 2 caption.

**Table 1 polymers-13-03445-t001:** Effect of different levels of EOs on radial growth and inhibition zone of *Aspergillus* sp.

EOs	Radial Growth (mm)				Inhibition Zone (mm)			
	0.5%	1%	2%	5%	0.5%	1%	2%	5%
Control	7.7 ± 0.4 ^a^	8.1 ± 0.6 ^a^	8.9 ± 0.6 ^a^	9.1 ± 0.3 ^a^	-	-	-	-
Lemongrass	6.7 ± 0.7 ^b^	6.1 ± 0.9 ^b^	5.5 ± 0.8 ^c^	5.4 ± 0.7 ^c^	0.4 ± 0.02 ^b^	0.8 ± 0.04 ^b^	1.3 ± 0.07 ^b^	1.4 ± 0.2 ^b^
Spearmint	7.8 ± 0.2 ^a^	6.2 ± 0.4 ^b^	6.3 ± 0.8 ^bc^	6.2 ± 0.6 ^b^	0.5 ± 0.01 ^b^	0.8 ± 0.03 ^b^	0.8 ± 0.02 ^c^	0.9 ± 0.04 ^c^
Ginger	6.9 ± 0.8 ^ab^	5.6 ± 0.9 ^bc^	6.1 ± 0.8 ^bc^	5.7 ± 0.9 ^c^	0.5 ± 0.03 ^b^	0.6 ± 0.03 ^b^	0.7 ± 0.09 ^c^	0.8 ± 0.02 ^c^
Bergamot	6.3 ± 0.9 ^b^	6.9 ± 0.6 ^b^	5.7 ± 0.7 ^c^	5.4 ± 0.8 ^c^	0.5 ± 0.01 ^b^	0.7 ± 0.02 ^b^	0.7 ± 0.02 ^c^	0.9 ± 0.05 ^c^
Lemon	7.5 ± 0.7 ^a^	7.7 ± 0.5 ^a^	7.5 ± 0.5 ^b^	6.9 ± 0.7 ^b^	0.4 ± 0.02 ^b^	0.6 ± 0.01 ^b^	0.6 ± 0.01 ^c^	0.6 ± 0.01 ^c^
Peppermint	7.6 ± 0.4 ^a^	7.7 ± 0.9 ^a^	7.4 ± 0.8 ^b^	6.7 ± 0.7 ^b^	0.3 ± 0.01 ^b^	0.7 ± 0.02 ^b^	0.8 ± 0.01 ^c^	0.8 ± 0.02 ^c^
Thyme	6.0 ± 0.8 ^b^	5.6 ± 0.9 ^bc^	4.1 ± 0.3 ^d^	3.1 ± 0.8 ^d^	0.4 ± 0.04 ^b^	0.7 ± 0.04 ^b^	1.4 ± 0.06 ^b^	1.6 ± 0.03 ^b^
Cinnamon	6.7 ± 0.3 ^b^	5.3 ± 0.8 ^bc^	4.1 ± 0.6 ^d^	3.3 ± 0.6 ^d^	0.5 ± 0.03 ^b^	0.8 ± 0.05 ^b^	1.1 ± 0.01 ^b^	1.5 ± 0.08 ^b^
Clove	5.2 ± 0.8 ^c^	4.7 ± 0.7 ^c^	2.5 ± 0.4 ^e^	1.3 ± 0.3 ^e^	0.9 ± 0.05 ^ab^	1.5 ± 0.04 ^a^	1.9 ± 0.05 ^a^	2.9 ± 0.09 ^a^

Values are mean ± standard deviation (n = 6). Different superscripts within the same column followed by different letters (a–e) indicate a significant difference (*p* < 0.05). Concentrations of different EOs are presented in µg/100 µL.

**Table 2 polymers-13-03445-t002:** Antifungal properties of EOs on *Aspergillus* sp. treated without and with different levels of EOs.

EOs	0.5%		1%		2%		5%	
	MIC	MFC	MIC	MFC	MIC	MFC	MIC	MFC
Lemongrass	25 ^b^	13 ^b^	25 ^c^	13 ^c^	13 ^c^	6 ^c^	6 ^d^	3 ^d^
Spearmint	100 ^a^	50 ^a^	50 ^b^	25 ^b^	25 ^b^	13 ^b^	25 ^b^	13 ^b^
Ginger	100 ^a^	50 ^a^	100 ^a^	50 ^a^	50 ^a^	25 ^a^	50 ^a^	25 ^a^
Bergamot	100 ^a^	50 ^a^	100 ^a^	50 ^a^	50 ^a^	25 ^a^	25 ^b^	13 ^b^
Lemon	100 ^a^	50 ^a^	100 ^a^	50 ^a^	50 ^a^	25 ^a^	50 ^a^	25 ^a^
Peppermint	100 ^a^	50 ^a^	50 ^b^	25 ^b^	13 ^c^	6 ^c^	13 ^c^	6 ^c^
Thyme	25 ^b^	13 ^b^	25 ^c^	13 ^c^	6 ^d^	3 ^d^	6 ^d^	3 ^d^
Cinnamon	100 ^a^	50 ^a^	25 ^c^	13 ^c^	6 ^d^	3 ^d^	6 ^d^	3 ^d^
Clove	25 ^b^	13 ^b^	13 ^d^	6 ^d^	6 ^d^	3 ^d^	3 ^e^	2 ^e^

Values are mean ± standard deviation (n = 6). Different superscripts within the same column followed by different letters (a–e) indicate a significant difference (*p* < 0.05). EOs: essential oils; MIC: minimum inhibitory concentration; MFC: minimum fungicidal concentration. MIC and MFC values present the concentration of EOs in µg/ 100 µL.

**Table 3 polymers-13-03445-t003:** Acceptance score of table grapes without and with antifungal packaging supplemented with 1% CEO at day 0 and day 21 of storage at 13 °C and RH 75%.

Samples	Storage Time (days)							
	Day 0				Day 21			
	Appearance	Firmness	Odor	Overall	Appearance	Firmness	Odor	Overall
Control	8.5 ± 0.2 ^aA^	8.6 ± 0.3 ^aA^	8.5 ± 0.2 ^aA^	8.5 ± 0.3 ^aA^	3.5 ± 0.1 ^aB^	3.8 ± 0.3 ^aB^	3.2 ± 0.2 ^aB^	3.5 ± 0.1 ^aB^
CEO1-Pad	8.6 ± 0.1 ^aA^	8.7 ± 0.1 ^aA^	8.6 ± 0.3 ^aA^	8.6 ± 0.2 ^aA^	4.4 ± 0.3 ^bB^	4.3 ± 0.2 ^bB^	4.1 ± 0.4 ^bB^	5.3 ± 0.3 ^bB^
COM-SO_2_-Pad	8.5 ± 0.2 ^aA^	8.6 ± 0.2 ^aA^	8.5 ± 0.3 ^aA^	8.5 ± 0.2 ^aA^	5.5 ± 0.2 ^cB^	5.7 ± 0.1 ^cB^	5.6 ± 0.3 ^cB^	5.6 ± 0.2 ^cB^
CEO1-HC-Film	8.6 ± 0.1 ^aA^	8.7 ± 0.3 ^aA^	8.7 ± 0.2 ^aA^	8.6 ± 0.1 ^aA^	6.3 ± 0.1 ^dB^	6.4 ± 0.2 ^dB^	6.1 ± 0.3 ^dB^	6.3 ± 0.2 ^dB^
CEO1-Film	8.8 ± 0.3 ^aA^	8.8 ± 0.2 ^aA^	8.8 ± 0.1 ^aA^	8.8 ± 0.3 ^aA^	7.2 ± 0.2 ^eB^	7.3 ± 0.1 ^eB^	7.2 ± 0.2 ^eB^	7.2 ± 0.1 ^eB^

Values are mean ± standard deviation (n = 50). Different uppercase and lowercase superscripts within the same row and same column indicate a significant difference (*p* < 0.05). TG: table grape; CEO: clove essential oil; Control: TG without any treatment; CEO1-Pad: TG with 1% CEO pad; COM-SO_2_-Pad: TG with commercial sulfur dioxide pad; CEO1-HC-Film: TG wrapped with a film composed of 1% CEO and halloysite clay; CEO1-Film: TG wrapped with 1% CEO film.

## Data Availability

The data presented in this study are available on request from the corresponding author.

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
