# Peer review of "Effect of Active Packaging Material Fortified with Clove Essential Oil on Fungal Growth and Post-Harvest Quality Changes in Table Grape during Cold Storage"

_polymers, 2021, doi:10.3390/polym13193445_

Round 1

Reviewer 1 Report

In general: I think the authors should analyze the chemical composition of CEO in days zero and 21. The increasing of the inhibition of EO should be discussed based upon the change of chemical composition especially the major components. 

In abstract:

  1. .......21 days at 13 °C and 75% RH.    Please RH is written here for the 1st time so it should be written as complete and the abbreviation in brackets
  2.  Lines 21-25: this introduction is 5 lines is so long. Please summarize

In Introduction

  1. The authors should described the scientific history of using of polyvinyl alcohol in this field if present.
  2. The authors should described in brief the history of clove as antifungal
  3. Figure 1: The authors should re-construct this figure with the same order of Tables 1 and 2 to be easier for any reader to compare

Author Response

******Authors are thankful for the noteworthy remarks and suggestions of reviewer#1 on our manuscript. Authors have revised the manuscript and conducted an English spell check and corrections were made while reading the whole manuscript between the lines. For methodology, frankly speaking, authors tried to reduce the details or methodology to avoid plagiarism. However, more information has been supplemented following the reviewer's insightful remarks. More details have been provided in the results as directed by the reviewer. Please see the corrections highlighted in yellow.  

Reviwer#1: Comments and Suggestions for Authors

In general: I think the authors should analyze the chemical composition of CEO in days zero and 21.

******Authors appreciate the insightful comment raised by reviewer#1. From the recent study of Kaur, Kaushal, and Rani (2019), GC-MS analysis of CEO identified the various components such as oxygenated monoterpenes (84.31%),  sesquiterpene hydrocarbons (15.45 %), oxygenated sesquiterpenes. The most abundant compounds were Eugenol (69.68 percent area), followed by Eugenyl acetate (14.38%) and β-Caryophyllene (12.23%). Authors do agree with the noteworthy comment about the analysis of CEO chemical composition at day 0 and day 21. However, authors have only focused on the antifungal effect of CEO under in vitro conditions and foremost on the application of CEO fortified polyynyl alcohol (PVA) film on the physical and sensorial quality of table grapes. Furthermore, the valuable suggestion of the reviewer will be employed in our next part of research that is under investigation related to the biochemical characteristics of table grapes as affected by the CEO antifungal packaging composite. The information related to the chemical composition of the CEO has been supplemented and cited in the revised manuscript (lines 73-79) and has been also referenced as follows:

Reference:-

Kaur, K., Kaushal, S., & Rani, R. (2019). Chemical Composition, Antioxidant and Antifungal Potential of Clove (Syzygium aromaticum) Essential Oil, its Major Compound and its Derivatives. Journal of Essential Oil Bearing Plants, 22(5), 1195-1217. doi:10.1080/0972060X.2019.1688689

In abstract:

  1. .......21 days at 13 °C and 75% RH. Please RH is written here for the 1st time so it should be written as complete and the abbreviation in brackets

*****Thanks for the noteworthy correction. Change has been made. Please see line (32)

  1. Lines 21-25: this introduction is 5 lines is so long. Please summarize

*****Thank you so much for raising an important point. Such irrelevant subjects have been removed from the abstract and have been rewritten to reflect the necessities of current research. Please see lines 21-22.

In Introduction

  1. The authors should described the scientific history of using of polyvinyl alcohol in this field if present.

      *****Thanks for the insightful comment. More information about the scientific history of polyvinyl alcohol has been provided in the revised manuscript. Please see the added information in lines 80-92.

  1. The authors should described in brief the history of clove as antifungal

*****Thanks for the valuable comment. Authors have provided a brief history of the antifungal efficacy of clove in the revised manuscript. Please see the highlighted line 76-79..

  1. Figure 1: The authors should re-construct this figure with the same order of Tables 1 and 2 to be easier for any reader to compare

*****Authors are thankful for the reviewer's suggestion about the reconstruction of ‘Figure 1’. Such a figure has been revised for better understand and readability for readers. Please see the revised ‘Figure 1’ in lines 288-291. Many thanks.

Reviewer 2 Report

In this manuscript the authors developed an active packaging embedded with essential oil to retarded the fungal growth and prolonged shelf-life of table grape during storage. This manuscript may be publishable, but some changes are needed.

The language needs some improvement.

Keywords:

- I suggest do not use the same words already used in the title.

Materials/Methods and Results:

- An improved explanation of why clove essential oil (1%) was selected as an active compound into packaging should be included.

- I don't see any data that are typical of packaging applications (such as mechanical properties, barrier, opacity).

- What is the release rate of the active compound from packaging?

- Are the results regarding active packaging promising? The authors should improve the final discussion, before it appears in the conclusion.

Conclusions: Should address more specific to the properties results presented in the manuscript as in the current form are too general. Also, should include the future-outlook.

Author Response

*****Thank you so much for your noteworthy remarks on our work. All the questions and suggestions have been responded to. English spell check was conducted in the whole manuscript. Results of the manuscript have been revised and more details of discussion have been provided as per the noteworthy evaluation of the reviewer. Please see all the corrections highlighted in green color.

Reviwer#2: Comments and Suggestions for Authors

In this manuscript the authors developed an active packaging embedded with essential oil to retarded the fungal growth and prolonged shelf-life of table grape during storage. This manuscript may be publishable, but some changes are needed. The language needs some improvement.

***** Authors also highly appreciate the reviewer’s insightful comments and understanding of our work. All the suggested changes have been made and highlighted in green color. English language of the manuscript has been improved and corrections have been made following insightful comments. Thanks.

Keywords:

- I suggest do not use the same words already used in the title.

*****Authors have revised the keywords following the reviewer's valuable suggestion. Please see the highlights lines 34-35. Thanks

Materials/Methods and Results:

- An improved explanation of why clove essential oil (1%) was selected as an active compound into packaging should be included.

*****Point raised by the reviewer has well taken into consideration for explaining the selection of clove essential oil in our work for fortification in antifungal packaging material. Firstly, the effects of different doses of all the EOs were analyzed under in vitro antifungal assays (radial growth, disk diffusion method, MIC, and MFC) of Aspergillus sp. Thereafter, preliminary research was conducted for the selection of specific doses of EOs compatible for order intensity score test in table grape berries. Grape berries were prepared in sterile plastic cups and 1 ml of each dose of all the EOs (0.5-5%) was pipetted out on the berries as detailed in the method (lines 146-156). Panelists were asked to evaluate the odor intensity score of each EO tested at different doses in grape berries. Berry samples treated with EOs at lower doses (0.5%) and higher doses (2-5%) were rejected due to very low and strong odor intensities that might affect the sensorial attributes of table grapes. Although EOs at higher doses efficiently retarded in vitro fungal growth. However, panelists analyzed a 1% dose of each EO was more compatible without any malodor developed in grape berries. Furthermore, berry samples with a 1% dose of each EO were evaluated to select the EO with a pleasant odor. 1% CEO was scored for the pleasant order in comparison with all the 1% EOs treated berry samples. Thus, the CEO at 1% was employed in the preparation of antifungal packaging material for safeguarding the quality of table grapes during storage. Authors have reported the important selected data for reporting ‘order intensity scores’ of all the 1% EOs for better understanding, otherwise, too much data may cause jumbling and difficulties for the readers. Please see the revised lines 168-169. Thanks.

- I don't see any data that are typical of packaging applications (such as mechanical properties, barrier, opacity).

*****Thanks for the noteworthy comment. Authors have only focused on the antifungal effect of CEO under in vitro conditions and foremost on the application of CEO fortified polyynyl alcohol (PVA) film on the physical and sensorial quality of table grapes. The insightful question of the reviewer about mechanical and barrier properties of PVA fortified CEO film will be reported in our next part of research that is under investigation that mainly will focus on physical (mechanical and barrier) properties of antifungal packaging and its impact on biochemical quality characteristics of table grapes. The authors highly appreciate the valuable point of the reviewer to generate more research data. 

- What is the release rate of the active compound from packaging?

*****Thanks for the insightful question. For the release rate of active compounds from packaging, authors want to conduct such a powerful technique. Unfortunately, we do not have the proper equipment for running experiments related to the active release of volatile compounds from packaging material. However, the suggestion by the reviewer has been taken into consideration for our next part of work in which the facility will be availed from the other research organization. Apologies and we do appreciate the invaluable suggestion.

- Are the results regarding active packaging promising? The authors should improve the final discussion before it appears in the conclusion.

***** Thanks for your insightful question and recommendation. As a matter of fact, results obtained in the table grapes packaged with CEO fortified PVA film was promising. It was noted that the active components released from the CEO inhibited fungal growth and prevented overall physical quality losses in table grape during storage. Authors have revised the final discussion specifically addressing the results of active packaging in detail. Please see the revised lines 217, 221-229, 234-243, 252-256, 277-279, 282-284, 354-355 highlighted in green color.

Conclusions: Should address more specific to the properties results presented in the manuscript as in the current form are too general. Also, should include the future-outlook.

*****Authors apologize for the unorganized structure of the ‘conclusion section’ of our manuscript. The content has been revised following the valuable comments of the reviewer. More specific and conclusive details have been provided reflecting the efficacy results of active packaging material on the physical, antifungal, and sensorial properties of table grapes. Please see the revised lines388-399. Thank you so much.

Round 2

Reviewer 1 Report

I think the authors did all the required revisions and I recommend the acceptance of the manuscript